# Platelets Contribution to Thrombin Generation in Philadelphia-Negative Myeloproliferative Neoplasms: The “Circulating Wound” Model

**DOI:** 10.3390/ijms222111343

**Published:** 2021-10-20

**Authors:** Alessandro Lucchesi, Roberta Napolitano, Maria Teresa Bochicchio, Giulio Giordano, Mariasanta Napolitano

**Affiliations:** 1Hematology Unit, IRCCS Istituto Scientifico Romagnolo per lo Studio dei Tumori (IRST) “Dino Amadori”, 47014 Meldola, Italy; alessandro.lucchesi@irst.emr.it; 2Biosciences Laboratory, IRCCS Istituto Scientifico Romagnolo per lo Studio dei Tumori (IRST) “Dino Amadori”, 47014 Meldola, Italy; teresa.bochicchio@irst.emr.it; 3Internal Medicine Division, Hematology Service, Regional Hospital “A. Cardarelli”, 86100 Campobasso, Italy; giuliogiordano@hotmail.com; 4Department of Health Promotion, Mother and Child Care, Internal Medicine and Medical Specialties and Infectious Disease Unit, University Hospital “P. Giaccone”, 90127 Palermo, Italy; mariasanta.napolitano@unipa.it

**Keywords:** MPN, platelet function, thrombin generation, PAR receptors, fibrinogen

## Abstract

Current cytoreductive and antithrombotic strategies in MPNs are mostly based on cell counts and on patient’s demographic and clinical history. Despite the numerous studies conducted on platelet function and on the role of plasma factors, an accurate and reliable method to dynamically quantify the hypercoagulability states of these conditions is not yet part of clinical practice. Starting from our experience, and after having sifted through the literature, we propose an in-depth narrative report on the contribution of the clonal platelets of MPNs—rich in tissue factor (TF)—in promoting a perpetual procoagulant mechanism. The whole process results in an unbalanced generation of thrombin and is self-maintained by Protease Activated Receptors (PARs). We chose to define this model as a “circulating wound”, as it indisputably links the coagulation, inflammation, and fibrotic progression of the disease, in analogy with what happens in some solid tumours. The platelet contribution to thrombin generation results in triggering a vicious circle supported by the PARs/TGF-beta axis. PAR antagonists could therefore be a good option for target therapy, both to contain the risk of vascular events and to slow the progression of the disease towards end-stage forms. Both the new and old strategies, however, will require tools capable of measuring procoagulant or prohaemorrhagic states in a more extensive and dynamic way to favour a less empirical management of MPNs and their potential clinical complications.

## 1. Introduction

Philadelphia-negative Myeloproliferative Neoplasms (MPNs) are diseases characterised by a high risk of thrombo-haemorrhagic events. In the past years, the focus has been on the mechanistic model of clonal proliferation, often attributing the states of hypercoagulability to the alteration of cell counts. First-line therapeutic strategies, almost always based on cytoreduction, aim at “numerical” rather than functional objectives. Thrombosis in MPNs can occur at any platelet count, therefore it is questionable regarding what the correct way is to monitor the efficacy of therapeutic strategies. Although the importance of cellular interactions has been demonstrated, in particular between white blood cells, platelets, and the vascular endothelium in JAK2-mutated patients, there are numerous experiences on the alteration of platelet function in MPN, which is partly secondary to a state of hypercoagulability, primarily plasma-driven, and has increased thrombin generation (TG) as its central element.

## 2. Platelets in MPNs: Role and Controversies

The role of platelets in the pathogenesis of vascular events in MPNs remains controversial [1,2]. Several studies did not observe a correlation between platelet count and thrombotic risk [3,4]. A recent review points out the correlation between the presence of driver mutations and platelet activation, underlining how this may actually depend on the consequences of the activation of myeloproliferative pathways on endothelial and systemic inflammation. It is also reiterated that cell counts have a substantially dubious role both on thrombotic risk (only in ET, leukocytosis appears to be a risk factor) and haemorrhagic risk (the majority of acquired von Willebrand diseases have a platelet count below 1000 × 10^9^ /L [5]. On the contrary, Michiels et al., described a platelet-mediated microvascular thrombotic syndrome, documenting thrombotic processes by a reduced platelet survival and by an increase in beta-thromboglobulin, platelet factor 4, and thrombomodulin levels in Polycythemia Vera (PV) and Essential Thrombocythemia (ET) patients [6]. A recent review published by Marin Oyarzùn and Heller accurately describes the platelet contribution to procoagulant states [7]. In ET and MF, a higher platelet activation rate has been observed [8,9] probably because of both intrinsic and extrinsic factors [7]. Among intrinsic platelet characteristics, we can discern membrane abnormalities [10] and JAK2-signalling hyperactivation. Endothelium can be also JAK2-mutated, releasing increased quantities of P-selectin and the von Willebrand Factor (vWF), and expressing higher levels of adhesion molecules and receptors, enhancing platelet activation [11]. The role of megakaryocytes in platelets activity has also been investigated, observing alterations in gene expression and the transcriptome [12,13]. One of the most recent and interesting findings is an altered expression of β-1,4-galactosyltransferase1 (B4GALT1) in megakaryocytes from MPN, which can induce the genesis of platelets with aberrant galactosylation [14]. Consequently, TPO synthesis by hepatocytes is promoted regardless of the circulating platelet mass. The expression of B4GALT1 can be modulated by JAK1/2 inhibitors, but the gene product should be considered a potential target for therapy.

Additionally, among extrinsic factors involved in platelet activation, we can discern the higher interaction with the endothelium and leukocytes, as well as the influence of miRNA-signalling and thrombin generation [7,15]. In fact, there is evidence of how driver mutations (and their allelic burden) may be responsible for a state of hypercoagulability induced both by a greater probability of interaction between platelets, leukocytes, and the endothelium, and by a more massive release of plasma factors [15,16]. Among these is the tissue factor (TF), which is an element whose function directly affects the generation of thrombin and whose circulating levels are particularly high in MPNs [17]. Often in clinical practice, it is thought to use hydroxyurea (and cytoreduction in general) with the aim of controlling cell counts, but probably the greatest advantage this strategy offers lies in the reduction of P-selectin-mediated TF expression in polymorphonuclear leukocytes [18]. The periodic assessment of cell counts may be a method too coarse to understand whether the patient is at risk for a cardiovascular event. Functional tests, such as global coagulation assays, which we will discuss later, would perhaps be more useful. In fact, these tests provide a reliable measure of the thrombogenic potential of thrombocytosis.

## 3. From Platelet Fibrinogen Receptors to Thrombin Generation: The “Circulating Wound” Model

The platelet fibrinogen receptor (PFR) is exposed only after a conformational change in glycoprotein (GP) IIb/IIIa, which is the most abundant integrin on the platelet surface. This process is exquisitely attributable to platelet activation, particularly in conditions of shear stress [19]. MPNs are generally characterised by hyper-viscosity phenomena, therefore the flow cytometric evaluation of the expression of the fibrinogen receptor should represent a refined method for the evaluation of platelet activation. As we will see in this section, the experimental data lead us to other considerations.

The active conformation of the CD41/CD61 complex (glycoprotein IIb/IIIa) is recognised by the PAC-1 antibody. In details, after vessel injury, CD41 (platelet glycoprotein IIb) interacts with CD61 (platelet glycoprotein IIIa) to form a functional receptor, exposing the fibrinogen-binding site and promoting platelet aggregation [20,21]. In MPNs, intrinsic cellular abnormalities in the GPIIb/IIIa complex activation have been described [22] and JAK2-mutated ET patients presented an altered functionality of the PI3 kinase/Rap1, with a reduced activation of the GPIIb/IIIa receptor [23]. Moreover, in ET, a decreased concentration of glycoprotein IIb and IIIa was observed, with a reduced sialysation and a consequent low fibrinogen binding [24,25,26,27].

A recent study by Marín Oyarzùn et al., focused on the role of platelets in immunity, inflammation, and thrombosis, showing an enhanced toll-like receptor 2-mediated translocation of granule membrane proteins with preserved GPIIb/IIIa activation [28]. A decrease in PAC-1-binding was also documented by Jensen et al., despite a normal expression of GPIIb/IIIa [22]. Finally, our recent work confirmed a reduced activation of GPIIb/IIIa in MPN patients, with a recovery of the PAC-1-binding capacity after acetylsalicylic acid intake [29]. The most interesting aspect of our experience was the recovery of PFR expression—to levels very close to that of healthy subjects—in patients under prophylaxis with low doses of acetylsalicylic acid. Paradoxically, in the context of diseases characterised by a state of hypercoagulability, we observed the presence of platelets unable to bind fibrinogen. The most credible explanation of this phenomenon, however, distances itself from an alleged “thrombasthenia”.

In fact, the experimental observations obtained by Moore in 2013 showed an altered binding of fibrinogen to platelets in patients with ET [23]. Although GPIIb/IIIa was normally expressed under baseline conditions, the authors observed a significant increase in its protease-activated Receptor-1 (PAR-1)-mediated exposure after stimulation with thrombopoietin. A rapid disappearance of the fibrinogen binding sites on the platelets followed soon after. We therefore speculated that a more marked thrombin generation in MPNs could be at the basis of an increase in PAR-1 activity, for instance, to determine a continuous conversion of functional fibrinogen into fibrin, as well as concerning the reduction of PFR. Acetylsalicylic acid (ASA), a drug also known for its fibrinolytic and hypoprothrombinemic properties, would thus be able to intervene on plasma factors and restore the ability to bind fibrinogen to platelets [30].

Protease-activated receptors (PARs) are protein-coupled receptors responsible for protease-signalling and for regulating cellular processes, survival, and apoptosis. They are expressed in platelets and the endothelium, with a role in haemostasis regulation [31,32]. Genetic polymorphisms of PAR-1, PAR2, and PAR-4 have been recently described [33] and they seem able to act on the gene-coding sequences and their degree of expression, or may regulate the downstream signalling. In particular for PAR-4, single nucleotide polymorphisms could influence the downstream response of PARs, even though their specific effects on platelet activation still need to be clarified [34].

The thrombin promotes platelet activation by the cleavage of the NH2-terminal domain of PAR-1 and PAR-4. Activated PAR-1 stimulates RhoA activation though ERK1/2 kinases [32,35,36], inhibits the accumulation of cAMP, activates phospholipase C, and promotes Ca2+ mobilisation [37,38]. Moreover, PAR-1 activation influences endothelial barrier permeability due to the stimulation of the MAPKs cascade [39]. PAR-4 exerts several functions similar to PAR-1 [40,41], as it presents a more massive calcium signal [42] and is involved in cellular blebbing due to RhoA and B-arrestin activation [43]. Finally, it promotes platelet activation through G protein-coupled receptor kinase 5 (GRK5) [44].

The PAR4 structure shows several differences from the other PARs, its extracellular amino-terminus and intracellular carboxy terminus has little sequence similarity to the corresponding regions of other PARs [45]. PAR-4 does not show a high affinity thrombin-binding domain, as found in the other thrombin receptors [46].

Several studies described a role of PARs in cancer development, in particular in the setting of pancreatic cancer, where PAR-1 appears to be crucial for disease progression, for promoting an immunosuppressive microenvironment, and for conferring chemoresistance [47,48,49,50,51]. PAR-1 germline polymorphisms are linked to the prognosis of the tumour [52]. On the contrary, we still know little about the pathogenetic role that PARs play in MPNs. However, if we start from the consideration that both pancreatic cancer and MPNs result in TGF-beta-mediated fibrosis in their advanced stage [53,54], we could be determinedin designing the next experiments. In a 2018 paper published by Ungefroren et al., in this Journal, the signalling crosstalk between the TGF-beta/ALK5 and PAR-2/PAR-1 pathways was accurately described [55]. In summary, in different models of disease ranging from cancer to simple wound healing, the activation of PARs generates platelet activation and the release of TGF-beta, which in turn regulates PAR-1 and PAR-2 at a traditional level, generating a vicious circle that is self-maintaining and pushes towards both fibrosis and tumour growth.

In conclusion, MPNs would therefore be a model that—starting from a TF-rich clonal platelet—feeds a “circulating wound” (Figure 1).

## 4. Thrombin Generation and Platelet Dysfunctions

Thrombin generation is a finely regulated process. After injury, the damaged endothelium exposes the tissue factor (TF), activating the coagulation cascade with thrombin production. Thrombin is responsible for the conversion of fibrinogen into fibrin, for the activation of several factors of the coagulation cascade (such as V, VIII, and XI), and for platelet recruitment and activation [56]. Platelets are activated through the cleavage of glycoprotein V and through the stimulation of PAR-1 and PAR-4 receptors [57]. The formed plug is then protected by the inhibition of ADAMTS13 activity [58]. Activated platelets present a procoagulant potential and can expose phosphatidylserine as a substrate for the conversion of prothrombin into thrombin [59]. MPNs showed the downregulation of several genes involved in thrombin generation [13], with a direct correlation between JAK2(V617F) allele burden and thrombin generation potential [60]. ET presented a higher production of platelet-induced thrombin in JAK2-mutated patients [61] and a recent study, performed in JAK2-mutated primary myelofibrotic mice models, documented a reduction in thrombus and size formation [62]. MF patients presented, indeed, a reduced endogenous thrombin potential, significantly correlated with platelet count [63].

Higher levels of platelet-released procoagulant microparticles, increasing thrombin generation in PV and ET patients, have also been detected [64,65], leading to a thrombomodulin resistance and probably contributing to the hypercoagulable state of patients [64]. For PV and ET patients, a reduced thrombin potential, if compared to healthy controls, has been described, also showing the occurrence of an acquired activated protein C resistance [66]. Thrombin generation has also been proposed as a potential biomarker of thrombotic risk in MPN [67].

Another aspect of considerable interest is the thrombin-mediated generation of intraceullary reactive oxygen species (ROS), which also require the activation of PAR-4 [68]. ROS overexpression is known to be linked to a proliferative advantage of mutated JAK2 clones and associated with an increased incidence of thrombotic events [69]. This is another excellent example of the interconnection between the hypercoagulability and inflammation in MPNs.

## 5. How to Evaluate Coagulation Parameters: Global Coagulation Assays

Patients with myeloproliferative neoplasms (MPN), even within the course of thrombosis, show little or no abnormalities of conventional coagulation tests. Conventional coagulation tests (CCTs) are not able to assay interactions between clotting factors, blood cell elements, and the vascular endothelium, thus they cannot predict and/or guide therapy in acute haemorrhages and are unable to predict thrombotic risk [70].

Global haemostatic assays (i.e., thrombin generation in platelet-rich plasma and thromboelastometry or thromboelastography in whole blood) have been evaluated for their role in detecting signs of procoagulant imbalance in patients with MPNs. Available data are, however, quite scant.

In 2013, Tripodi et al., evaluated thrombin generation in platelet-rich plasma and thromboelastometry in 111 patients with MPN and in 89 controls [71]. The endogenous thrombin potential (ETP) revealed to be higher in patients than in controls. ETP directly correlated with platelet counts, while there was an inverse correlation with free protein S, protein C, and antithrombin. Patients under hydroxyurea treatment showed lower ETP ratios than those on other treatments.

In 2016, Giaccherini et al., performed a study on ROTEM using INTEM and EXTEM reagents in patients with ET (*n* = 39) and PV (*n* = 23), while nineteen healthy subjects acted as the controls [72]. The ROTEM analysis showed a hypercoagulable state in MPN patients, with shorter CFT and higher MCF in comparison to the controls, with both reagents. A statistically significant correlation was found between platelet count and MCF or CFT. Platelet count resulted independently as associated to ROTEM parameters at the multivariate analysis. MCF values, corrected for the platelet count, revealed a lower platelet reactivity in the enrolled patients.

Plasma samples from 36 patients with MF (JAK2 V617-positive, 53%; CALR-positive, 31%; MPL-positive, 14%; and triple negative, 2%) and from 20 healthy volunteer blood donors were assayed by Tthrombin generation in PRP and in a fully automated system. Results were analysed for their correlation with clinical and laboratory parameters of the enrolled patients. Differences in ETP between the two groups were found, as ETP was lower in the patient group (*p* = 0.0003). Multivariate analysis confirmed a significant correlation between thrombin generation and platelet counts, with higher thrombin generation in patients with thrombocythemia >400 × 10^9^/L (*p* = 0.04). ETP was higher in earlier stages of MF and lower in CALR-mutated samples. Authors concluded that thrombin generation in MPNs is mainly influenced by platelet counts and thrombocythemia may be a potential risk factor for thrombotic complications [63].

In another study, thirty-eight patients with MPN were enrolled (median age: 65 years), including 26 patients with essential thrombocythemia (68.4%), eight with PV (20.5%), three with MF, and one with unclassifiable MPN; blood samples were evaluated by thromboelastography and thrombin generation (CAT) [73]. Compared with the controls, there was no difference in the maximum amplitude and lysis time (LY30) was significantly higher in the thromboelastography. The CAT showed a higher thrombin peak and velocity index with comparable ETP. Fibrin generation parameters were significantly reduced with preserved overall fibrinolytic potential and P-selectin was markedly increased. This study showed specific differences between subjects with MPN and normal controls. A high lysis time (LY30) with reduced fibrin generation in MPN patients apparently did not fit with the prothrombotic characteristics of MPN, probably because they may mirror a compensatory attempt to balance haemostasis. It is most likely that the study included large percentages of patients on antiplatelet prophylaxis (95% of subjects) and/or undergoing cytoreductive therapy (60%). As already mentioned, the contribution of these treatments in promoting hyperfibrinolysis and dampening thrombin generation is widely conceivable. A small number of patients were receiving phlebotomies but we do not know the contribution of this strategy to the modification of the thromboelastographic parameters.

The circulating procoagulant activity (CPA) of microparticles in polycythaemia vera (PV) and in essential thrombocythemia (ET) has also been determined by a thrombin generation test performed in the absence and presence of thrombomodulin (TM), wherein TM-resistance was observed and postulated to contribute to the hypercoagulable state of MPN [74].

Sain et al., evaluated the role of Rotational Thromboelastometry (ROTEM) to screen the coagulation profile of patients with MPNs. Authors found higher mean maximum clot firmness values in patients affected by ET and PV compared to the healthy controls [75]. The authors highlighted how thromboelastography is able to finely detect those states of the hypercoagulability characteristic of MPNs. Furthermore, since clotting time is more influenced by coagulation factors rather than by platelet hyperaggregability, the hypothesis of a greater plasma contribution is supported. This implies the potential usefulness of tests such as ROTEM to dynamically measure global coagulation and to be combined with routine cell counts.

## 6. Current Therapies and Rising Therapeutic Alternatives

Low-dose ASA is the antithrombotic strategy that can be commonly used in MPN, thus making use of it is probably excessive compared to the real needs [76,77,78,79]. While the advantage in the management of PV is evident [80], in other pathological contexts such as MF or ET (especially in the presence of a CALR mutation), antiplatelet prophylaxis is at least questionable also because of the possible bleeding complications [76,77,78,79]. In ET patients receiving ASA, the increments in f-MLP-induced PMN-CD11b and in PMN-platelet aggregates were significantly lower versus the ET subjects not treated with ASA [81]. Resistance phenomena are not infrequent, often manifesting themselves with microcirculation disorders, which can be overcome with the administration of low doses twice a day (more effective than doubling the dosage by maintaining a single administration) [82,83,84]. In our study on platelet fibrinogen receptor expression in MPNs, by focusing on the group of patients under ASA and with no history of thrombosis, it was found that subjects with persistent microcirculatory disorders show a higher PAC-1-binding capacity if compared to the asymptomatic ones [29]. As said before, the fibrinolytic and hypoprothrombinemic properties could be more important than the pure antiplatelet properties; for this reason, we have often wondered if ASA is really to be considered the best strategy in MPNs.

Hydroxyurea remains the most used drug in cytoreduction. Long-term experiences with this compound show excellent safety characteristics [76]. Compared to other drugs, moreover, it can dampen the generation of thrombin [85]. To determine the efficacy of the treatment, the determination of the platelet count is used, as it seems to be a good surrogate for TG [63]. However, it could be important to introduce periodic functional tests during treatment, starting with studies that evaluate their usefulness in the long term, given the incidence of vascular events in patients under combined treatment (cytoreduction in addition to antiplatelet or anticoagulant prophylaxis). The phenomenon of resistance or intolerance to hydroxyurea also opens the problem of the absence of similarly effective strategies for controlling TG [86].

The precision therapies currently available in MPNs—namely JAK inhibitors—are capable of exerting effects on GP-VI-mediated platelet functions but have no influence on the platelet response to thrombin. This evidence also confirms the need for new and targeted treatment options [87].

Theoretically, an anti-PAR molecule could offer more complete coverage, targeting the hypercoagulability state induced by impaired thrombin generation, as well as the inflammation and promotion of fibrosis resulting from the platelet activation and crosstalk of the previously mentioned TGF-beta/ALK5 and PAR-2/PAR-1 pathways [88,89,90,91,92,93,94,95]. Such an approach could perhaps offer a contribution to slowing the evolution of the disease. However, the only drug of this class approved by the FDA for the prevention of cardiovascular events—the anti-PAR-1 Vorapaxar—seems to have clinical use limits linked to the risk of bleeding, including intracranial bleeding in patients with previous stroke [96,97,98]. Several authors have suggested that translational and clinical research be focused on studying anti-PAR-4, also in consideration of the PAR-4 multifaceted role [98,99,100]. Numerous anti-PAR-4 inhibitors have been synthetised but few have been considered for clinical trials, probably due to the difficult selection of their targets [101]. However, anti-PAR4 antibodies have been recently experimentally evaluated for their targeted action on the thrombin cleavage site of PAR4, showing, in vitro, a specific and effective action on a human thrombosis model. Among them, we found function-blocking antibodies, peptidomimetics, low molecular weight compounds, and pepducins [99]. From a preclinical perspective, the combination of THE PAR-4 and PAR-1 inhibitors has been evaluated, showing promising results [102].

Dabigatran also acts as a PAR-1 inhibitor, combining anticoagulant and antiplatelet effects [103,104]. Among the direct oral anticoagulants (DOACs), this molecule could be the subject of specific studies, even if burdened by a higher rate of haemorrhagic events in MPNs [105]. It should be noted that the Dabigatran-dependent thrombin inhibition is useful in enhancing the inhibition of the growth and dissemination of pancreatic tumour cells by means of a synergistic effect with gemcitabine (the use in monotherapy seems instead deleterious) [106]. The safety data of DOACs, with the limitations of a retrospective study, are comparable to those of vitamin K inhibitors and these drugs appear to effectively protect against the risk of thrombotic recurrence. In patients with atrial fibrillation and MPN, the low rate of cerebrovascular ischemic events may, however, be conditioned by the concomitant use of hydroxyurea [105].

## 7. Conclusions

Platelets seem to significantly contribute to the pathogenesis of thrombosis within the course of MPNs. Here, we have reviewed available molecular evidence on the role of platelets, gene polymorphisms, and receptors in contributing to hypercoagulability and sustaining thrombosis. We focused on a particular pathogenic mechanism as we believe that the platelets produced by the clonal process are responsible for the increased thrombin generation and that the consequent perpetual activation of the TGF-beta/ALK5 and PAR-2/PAR-1 pathways is a central element both for the risk of vascular events and for the progression of the disease.

Here, we propose a new conceptual model, which we believe could help in identifying the next goals. In fact, even though we have significantly improved our knowledge regarding the pathogenesis, risk stratification, and effective biologic treatments of Ph-neg MPNs, arterial thrombosis and venous thromboembolism (VTE) still represent the main reasons for the morbidity and mortality of patients with MPNs [107], thus an accurate and sensitive laboratory evaluation of the risk of thrombosis onset and recurrence may positively impact the management of MPNs.

The biological complexity behind the thrombotic risk in MPNs is evidenced by the numerous potential biomarkers that can be used in the context of a liquid biopsy [108]. Our model, therefore, does not claim to be exhaustive with respect to the various mechanisms underlying hypercoagulability, but it aims to interconnect a part of them and to highlight sensitive elements for targeted therapeutic strategies. A tailored approach to MPN-related thrombosis should help to prevent bleeding complications during long-term anticoagulant treatment. Molecular therapeutic targets interfering with platelet activation may be effective in the primary prophylaxis of VTE in patients with MPNs. Furthermore, treating coagulation disorders in MPNs in a less empirical way could have an important impact on other pathogenetic aspects and on the transformation of these diseases. The interconnection described between coagulation and fibrosis is a clear example of this. Prospective ad hoc studies based on a combined molecular and clinical approach should be designed to answer a still open “wound” in MPNs.

## Figures and Tables

**Figure 1 ijms-22-11343-f001:**
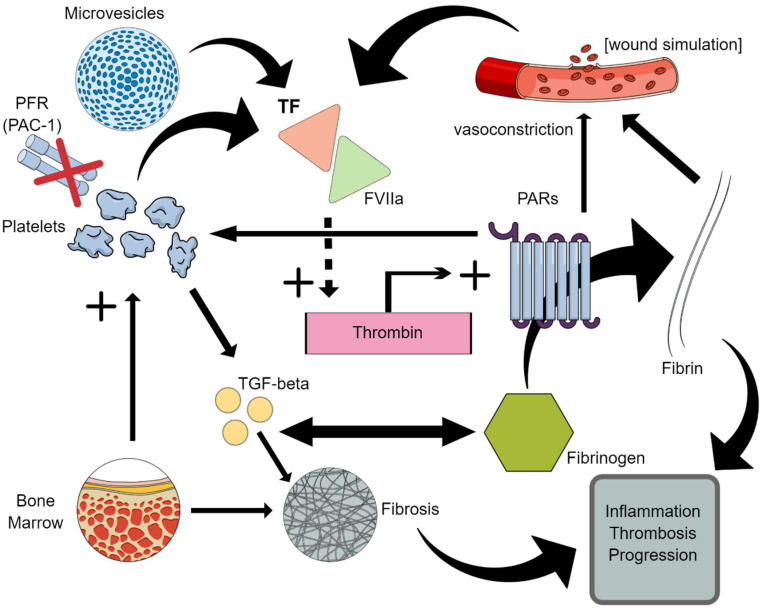
The “circulating wound” model. Clonal platelets and microvesicles generate an excessive release of tissue factor (TF), which occurs after tissue and vascular lesions. The triggering of the coagulation cascade results in an increased generation of thrombin, which acts as a protease and activates protease-activated receptors (PARs). PARs perform multiple functions: they rapidly convert functional fibrinogen into fibrin (by making the fibrinogen binding sites on platelets disappear, PFR), promote vasoconstriction, and ensure the platelets are activated. Signalling crosstalk with the TGF-beta pathway contributes to the genesis of inflammation and fibrosis.

## Data Availability

No new data were created or analysed in this study. Data sharing is not applicable to this article.

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
