# Peer review of "Platelets Contribution to Thrombin Generation in Philadelphia-Negative Myeloproliferative Neoplasms: The “Circulating Wound” Model"

_ijms, 2021, doi:10.3390/ijms222111343_

Round 1

Reviewer 1 Report

The authors reviewed the contribution of clonal platelets of MPNs in promoting an unbalanced generation of thrombin by tissue factor and protease activated receptors to define a "circulating wound” model, which advocated for increase activation of coagulation, inflammation and fibrotic progression. 

Major Comments

The systematic evaluation of the role of blood cells in the pathogenesis of thrombosis of MPNs has been recently published (Leukemia 2021;35(4):935-955. doi: 10.1038/s41375-021-01170-z; Curr Opin Hematol 2021 Sep 1;28(5):285-291. doi: 10.1097/MOH.0000000000000664; Curr Hematol Malig Rep 2021 Jun;16(3):304-313. doi: 10.1007/s11899-021-00630-8; Blood 2021 Apr 22;137(16):2152-2160. doi: 10.1182/blood.2020008109). Certain aspects of the manuscript deserve the following comments: 

  • The nomenclature has been changed from "chronic myeloproliferative diseases" to "myeloproliferative neoplasms" (MPNs) (2008 revision of the World Health Organization). Please correct in the text. 
  • Since the cause of thrombosis in patients with MPNs is multifactorial, the hypothesis of the authors, although attractive and original, is very simple (that is, the attribution to unique mechanism for thrombosis is biological unlikely). Please discuss. 
  • The authors concluded that “MPNs would therefore be a model that - starting from a TF-rich clonal platelet – feeds a “circulating wound”. Expression of TF by platelets and neutrophils is controversial, with some authors reporting expression of TF in these cells, whereas others show no evidence. In particular, there are contradictory reports about the TF expression in ET platelets (Haematologica. 2006;91:169, #59 and #60). The authors should discuss this aspect in their manuscript.
  • In support to authors' hypothesis, the increased of circulating TF is well known in MNPs (Am J Hematol 2009 Feb;84(2):102-8. doi: 10.1002/ajh.21338, Oncol Lett 2017 Aug;14(2):2531-2536. doi: 10.3892/ol.2017.6459, Medicina (Kaunas) 2019 Feb 16;55(2):54. doi: 10.3390/medicina55020054). 
  • The authors do not discuss two important issues for their general hypothesis, such as the study of JAK2 animal models in platelet function (Blood. 2013;122:3787–97 and Blood. 2014; 124:1136–45) and platelet turnover (Am J Hematol 2009 Feb;84(2):102-8. doi: 10.1002/ajh.21338, Blood 2011 Sep 1;118(9):2599-601. doi: 10.1182/blood-2011-02-339655Int J Hematol 2014 Nov;100(5):429-36. doi: 10.1007/s12185-014-1673-0).
  • It is difficult to understand how peak and slope of TG correlates with platelet counts when the TG test was performed at adjusted platelet counts (150,000 plts/µl) as in the reference #59. 
  • The assumption that “Aspirin, a drug also known for its fibrinolytic and hypoprothrombinemic properties,” is not supported by any reference. 
  • The point 5 (How to evaluate coagulation parameters: global coagulation assays) do not support the principal hypothesis of the authors.
  • Please delete the paragraph (pp 247-252) of page 6. 
  • The use of aspirin “twice a day” (page 6, 272) is not recommended in clinical practice due to lack of strong evidence (Blood Cancer J 2019;9:61).  
  • The authors do not include any result about “…. available molecular evidence on the role of platelets, gene polymorphisms …” (Conclusion). It is suggested to add literature in this regard (e. g. Balkan J Med Genet 2017 Jun 30;20(1):35-42. doi: 10.1515/bjmg-2017-0005). 
  • What are the authors referring to when they write ... “will require tools capable of measuring procoagulant or prohemorrhagic states” or “an accurate and sensitive laboratory evaluation of the risk of thrombosis”? To the need for a new method or to the use of global coagulation assays with scarce evidence? 
  • The manuscript will be benefited of the use a more brief and centered in the general hypothesis.

Minor comment

  • Background: “…necessary to determine this. in the…” should be “…necessary to determine this in the…” (page 7, pp 285). 

Author Response

We warmly thank the reviewer for this detailed analysis of the content of our manuscript. We have accepted some suggestions and provided extensive explanations for other additions which we believe are not completely consistent with the subject matter. We would like to be consistent with our communicative intentions.

Q1: The systematic evaluation of the role of blood cells in the pathogenesis of thrombosis of MPNs has been recently published (Leukemia 2021;35(4):935-955. doi: 10.1038/s41375-021-01170-z; Curr Opin Hematol 2021 Sep 1;28(5):285-291. doi: 10.1097/MOH.0000000000000664; Curr Hematol Malig Rep 2021 Jun;16(3):304-313. doi: 10.1007/s11899-021-00630-8; Blood 2021 Apr 22;137(16):2152-2160. doi: 10.1182/blood.2020008109). 

A1: Thank you for suggesting this review, which is of further support to our model. Paragraph 2 has been integrated with the relevant information.

Q2: The nomenclature has been changed from "chronic myeloproliferative diseases" to "myeloproliferative neoplasms" (MPNs) (2008 revision of the World Health Organization). Please correct in the text. 

A: We found no reference to the old nomenclature in the manuscript.

Q3: Since the cause of thrombosis in patients with MPNs is multifactorial, the hypothesis of the authors, although attractive and original, is very simple (that is, the attribution to unique mechanism for thrombosis is biological unlikely). Please discuss. 

A: Thanks for this discussion point. With due respect, we do not believe that the model we propose is an oversimplification of reality. Rather, we believe it is a good attempt to interconnect several mechanisms that are undoubtedly involved in the pathogenesis of thrombosis: the particular phenotype of activated platelet, the contribution of some plasma factors, the increasing levels of inflammation and fibrosis in the course of disease progression. Each of these elements can certainly be further detailed, but we are convinced that a good review should offer to the reader new theoretical-practical models on which to articulate the future translational research. The "circulating wound" offers another point of view with respect to the existing theories, which are divided between analogy and arithmetic mechanism. Our perspective finds its basis in the identification of consequential relationships, and in refined preclinical experiences. However, we understand and respect the intent of your objection, and we have integrated a strong sentence at the end of the article, to leave room for complementary interpretations.

Q4/Q5: The authors concluded that “MPNs would therefore be a model that - starting from a TF-rich clonal platelet – feeds a “circulating wound”. Expression of TF by platelets and neutrophils is controversial, with some authors reporting expression of TF in these cells, whereas others show no evidence. In particular, there are contradictory reports about the TF expression in ET platelets (Haematologica. 2006;91:169, #59 and #60). The authors should discuss this aspect in their manuscript. In support to authors' hypothesis, the increased of circulating TF is well known in MNPs (Am J Hematol 2009 Feb;84(2):102-8. doi: 10.1002/ajh.21338, Oncol Lett 2017 Aug;14(2):2531-2536. doi: 10.3892/ol.2017.6459, Medicina (Kaunas) 2019 Feb 16;55(2):54. doi: 10.3390/medicina55020054). 

A: In this case, the answer is contained in the question itself. We cannot compare the expression of mTF with the increase in circulating TF. Clearly, they are two different concepts, and the second is the reason for the impaired generation of thrombin. We found no references in the text to the expression of TF on the monocyte membrane (except in reference to a pleiotropic effect of hydroxyurea in the introductory part). If the reviewer agrees, we would not comment further on this point.

Q6: The authors do not discuss two important issues for their general hypothesis, such as the study of JAK2 animal models in platelet function (Blood. 2013;122:3787–97 and Blood. 2014; 124:1136–45) and platelet turnover (Am J Hematol 2009 Feb;84(2):102-8. doi: 10.1002/ajh.21338, Blood 2011 Sep 1;118(9):2599-601. doi: 10.1182/blood-2011-02-339655Int J Hematol 2014 Nov;100(5):429-36. doi: 10.1007/s12185-014-1673-0).

A: We have considered extending the discussion based on the suggestions, but the arguments are difficult to integrate without altering the "narrative" intent of the review, as they are conceptually very distant. The suggested references deal respectively with:
- genetics, formation of proplatelet and in vitro platelet aggregation of mouse models, an outdated experiment if compared with what is now known on ex vivo studies on MKC from humans in single cell sequencing, and in any case not relevant to the discussion;
- general aspects of in vitro haemostasis of JAK2V617F KI mouse models, with a special focus on pro-haemorrhagic features such as a decrease of HMW multimers of vWF and GPVI deficiency (which is overcome by thrombin): not relevant to the discussion;
- multiple elements (receptors, circulating / soluble factors, mutations) at the basis of some potentially prothrombotic alterations in MPNs: among these the high circulating levels of TF in JAK2 positive patients, which on the basis of this report we have reiterated in the text;
- two studies on immature platelets, on their thrombogenic potential and on the susceptibility to cytitoruductive treatment, not relevant to the discussion.

Q7: It is difficult to understand how peak and slope of TG correlates with platelet counts when the TG test was performed at adjusted platelet counts (150,000 plts/µl) as in the reference #59.

A: Thanks for letting us know. Since the information was not included in our review, we suggest asking the question to the authors of that article.

Q8: The assumption that “Aspirin, a drug also known for its fibrinolytic and hypoprothrombinemic properties,” is not supported by any reference. 

A: Thank you for reporting the involuntary omission. We took for granted an already dated scientific acquisition, but we managed to recover an article from 1989, which we have included among the references.

Q9: The point 5 (How to evaluate coagulation parameters: global coagulation assays) do not support the principal hypothesis of the authors.

A: The reviewer does not explain to us what the problem is, but we have included a sentence at the end of paragraph 2, which should clarify our position: both the platelet count and functional tests should be evaluated during treatment (they are not mutually exclusive).

Q10: Please delete the paragraph (pp 247-252) of page 6. 

A: Deleted - although it could be informative - because it refers to another myeloproliferative disorder.

Q11: The use of aspirin “twice a day” (page 6, 272) is not recommended in clinical practice due to lack of strong evidence (Blood Cancer J 2019;9:61).

A: The reviewer focuses on a much-debated topic and for which the evidence has never been conclusive. Thanks for the opportunity to discuss this issue. Actually, the suggested article (which has already been taken into consideration during the drafting), leads to very conclusions that differ completely from those reported in the reviewer’s comment. The current expert opionion recommend the use of aspirin BID, and the most recent evidence supports it whenever microcirculatory disturbances are not sufficiently controlled. We therefore believe that we cannot change this part. See also:

https://doi.org/10.1182/blood.2019004596

DOI: 10.1055/a-1334-3259

Q12: The authors do not include any result about “…. available molecular evidence on the role of platelets, gene polymorphisms …” (Conclusion). It is suggested to add literature in this regard (e. g. Balkan J Med Genet 2017 Jun 30;20(1):35-42. doi: 10.1515/bjmg-2017-0005). 

A: Thanks for pointing out a certainly interesting article, but the polymorphisms to which our conclusions refer are obviously those widely discussed in the text, regarding PAR receptors. With all due respect, we don’t feel like referring to "decontextualized" literature.

Q13: What are the authors referring to when they write ... “will require tools capable of measuring procoagulant or prohemorrhagic states” or “an accurate and sensitive laboratory evaluation of the risk of thrombosis”? To the need for a new method or to the use of global coagulation assays with scarce evidence?

A: The sentence reported is part of the abstract, while in the article the explanation of this statement takes up an entire paragraph. The latter contains all the evidence (increasing and of good quality) that the reviewer arbitrarily considers "scarce". We interpret this question as the possible persistence of an initial draft of the revision within the text which was then sent to our attention. This interpretation would also explain the next question.

Q14: The manuscript will be benefited of the use a more brief and centered in the general hypothesis.

A: [request not understandable]

Q15: Background: “…necessary to determine this. in the…” should be “…necessary to determine this in the…” (page 7, pp 285).

A: Our opinion is that the sentence would lose its meaning.

Reviewer 2 Report

The authors have beautifully reviewed the role of platelets in the generation of thrombin in Ph1-negative MPNs. The paper is well-written and worthy of publication. I have the following suggestions for revision:

  1. It would be better to refer to aspirin as acetylsalicylic acid as aspirin is a brand name.
  2. The references need to be revised to match the style of the journal. 
  3. A recent systematic review has pointed out that liquid biopsy and liquid biopsy biomarkers can be employed in the prediction of thrombosis in MPNs. See: https://www.mdpi.com/2075-1729/11/7/677
  4. Another contributor to the development of thrombosis in MPNs seems to be oxidative stress which works together with low-grade chronic inflammation in a vicious cycle in which reactive oxygen species stimulate the production of pro-inflammatory cytokines which in turn facilitate the generation of reactive oxygen species. See: https://doi.org/10.37358/RC.19.10.7581 & https://doi.org/10.37358/RC.19.8.7435

Author Response

We thank the reviewer for the support provided to this article, and for the valuable suggestions.

Q1: It would be better to refer to aspirin as acetylsalicylic acid as aspirin is a brand name.

A: The observation is totally acceptable; we have made the changes.

Q2: The references need to be revised to match the style of the journal.

A: Thanks for pointing out this problem. We have adjusted the references as suggested.

Q3/Q4: A recent systematic review has pointed out that liquid biopsy and liquid biopsy biomarkers can be employed in the prediction of thrombosis in MPNs. See: https://www.mdpi.com/2075-1729/11/7/677. Another contributor to the development of thrombosis in MPNs seems to be oxidative stress which works together with low-grade chronic inflammation in a vicious cycle in which reactive oxygen species stimulate the production of pro-inflammatory cytokines which in turn facilitate the generation of reactive oxygen species. See: https://doi.org/10.37358/RC.19.10.7581 & https://doi.org/10.37358/RC.19.8.7435

A: We believe the advice provided is excellent: by mentioning the liquid biopsy in the conclusions we made the final message even stronger, while the ROS found the right place at the end of the paragraph on the TG. We have thus integrated the content of two of the three references in the manuscript.

Round 2

Reviewer 1 Report

The authors have partially responded to some of the major concerns raised and the revised manuscript is improved. 

This reviewer has remaining issues for modification:

  • The use of the term “chronic” in myeloproliferative neoplasms is obsolete according to the current WHO classification.
  • Please check the phrase : “…necessary to determine this. in the…” (page 7, pp 285). 

Author Response

We thank the reviewer for having positively evaluated the corrections and additions made to the article.
We agree with the omission of the adjective "chronic" (given the more recent WHO definition). We have also rephrased the sentence which was definitely unclear. Thanks for pointing it out.